# Measuring Streambank Erosion: A Comparison of Erosion Pins, Total Station, and Terrestrial Laser Scanner

**Daniel T. Myers *** **, Richard R. Rediske and James N. McNair**

Annis Water Resources Institute, Grand Valley State University, 740 W. Shoreline Dr., Muskegon, MI 49441, USA
* Correspondence: myersda@mail.gvsu.edu; Tel.: +1-616-331-3749

**Abstract:** Streambank erosion is difficult to quantify; models and field methods are needed to assess this important sediment source to streams. Our objectives were to (1) evaluate and compare three techniques for quantifying streambank erosion: erosion pins, total station, and laser scanning, (2) spatially assess streambank erosion rates in the Indian Mill Creek watershed of Michigan, USA, and (3) relate results with modeling of nonpoint source pollution. We found large absolute and relative errors between the different measurement techniques. However, we were unable to determine any statistically significant differences between techniques and only observed a correlation between total station and laser scanner. This suggests that the three methods have limited comparability and differences between measurements were largely not systemic. Further, the application of each technique should be dependent on site conditions, project goals, desired resolution, and resources. The laser scanner collected high-resolution data on clear, barren streambanks, but the erosion pin and total station were more representative of complex vegetated banks. Streambank erosion rates varied throughout the watershed and were influenced by fluvial processes. We estimate that streambank erosion contributed 28.5% of the creek's total sediment load. These findings are important to address sources of watershed impairments related to sedimentation, as choosing an applicable technique for individual purposes can help reduce the challenges and costs of a streambank erosion study.

**Keywords:** streambank; erosion; lidar; stream; sediment; watershed

## 1. Introduction

Sediment pollution is a major concern for streams throughout the United States [1]. It causes widespread degradation of aquatic habitat and reduces suitability for fish and macroinvertebrate communities [1–3]. Sediment can enter a stream through many pathways, but the dominant pathway is often streambank erosion [4,5]. Streambank erosion is natural but can be accelerated by disturbances of changing watershed land use [1,2,6]. Successful management of sediment in a watershed requires an understanding of sources and entry pathways [7]. Understanding the dynamic nature of streambanks is important to shoreline landowners threatened by retreating banks, water quality managers, and geomorphologists [8]. It is also important for projects involving stream restoration and Total Maximum Daily Load development [9]. One difficulty with managing sediment pollution is that it is hard to quantify total sediment loads derived from streambank erosion [4,10]. Various techniques could be used for this purpose including erosion pins, total station surveying, and terrestrial laser scanning, however a comparison study of all three methods had not been performed over multiple streambanks with varying conditions. Thus, there were questions about the comparability of the techniques and their usefulness in different bank conditions. Our research explores this information gap by examining the performance of erosion pins, total station surveying, and terrestrial laser scanning under varying bank conditions.

## 1.1. Streambank Erosion Measurement Techniques

Erosion pins are narrow metal rods installed horizontally that are commonly used to measure the retreat of the streambanks over time [5,11,12]. They are suitable for a wide range of fluvial environments, inexpensive, and simple to maintain [11]. However, erosion pins can have difficulty accounting for spatial variability on streambanks [11]. They can also inflate erosion estimates because of bank destabilization during installation or turbulence caused by the pin [11].

A total station is an electronic surveying instrument that combines horizontal angle, vertical angle, and distance measurement [9,13]. Total station surveys can effectively show how the shape of a streambank changes over time from erosion or deposition by accurately measuring the locations of specific points [9,13]. However, data can be coarse (depending on the resources of the field survey) and lack point density needed to accurately model bank retreat and conditions [14]. Data collection with a total station can disturb the streambank [9]. Additionally, overhanging or undercut banks can make total station surveys difficult. There are no standard methods to account for the empty space below the overhang on digital elevation models. Undercut banks have previously been ignored because of this difficulty, which introduces a bias because data are only collected from certain forms of streambank [13]. This is important because streams can experience a channel widening stage from increased storm flows and water velocities [2], which could cause more undercut banks.

A terrestrial laser scanner uses lidar technology to create high-resolution point clouds of a surface showing three-dimensional topography by combining laser-based distance measurements with precise orientation [9,15,16]. A main advantage of terrestrial lidar is that it can detect minute changes in surface position and shape along a streambank, bluff, or gully with up to one millimeter resolution [17–19]. This allows managers to better monitor sediment sources spread over a channel network [17]. Though the technique provides superior measurement precision and accuracy, optical issues with water reflection [20] and obstruction by vegetation and crenulated surfaces [17,21,22] can interfere with measurement. Vegetation and other obstructions can be removed by programs that classify the point cloud data from a terrestrial laser scanner into different classes. However, the complexity of natural surfaces and size of data files makes vegetation classification difficult [22]. Heritage and Hetherington [21] recommend a field protocol for using a terrestrial laser scanner to study fluvial morphology. This includes positioning the scanner to minimize the shadowing of obstructions, placing targets for alignment in all three dimensions, and repeating scans from the same positions.

## 1.2. Prior Comparison Studies

Previous comparisons between techniques to measure streambank erosion have provided valuable insights into difference and error. Resop and Hession [9] compared a total station and terrestrial laser scanner for measuring streambank erosion along an 11 m streambank of Stroubles Creek, Virginia, USA with six readings over two years. The bank was bare, with little vegetation. Estimates of bank retreat rate were 0.15 m year$^{-1}$ with the laser scanner and 0.18 m year$^{-1}$ with the total station, thus a relative error of 20%. They found that the laser scanner was quicker to use and did not disturb the streambank like the total station. By comparing data points between the two methods, they found a mean bank retreat difference of 0.018 m, standard deviation of 0.020 m, and that 63% of total station points were within 0.02 m of the laser scanner data. Estimates of volumes of soil erosion from streambanks between the two techniques had an average difference of 109%, with a range from 7% to 373%. The cause of these differences was likely because of the different resolutions of the total station and laser scanner. Aside from some instances where an undercut bank clearly affected total station data, Resop and Hession did not find any systematic differences between the results of the total station and laser scanner on their bare bank.

Day et al. [23] compared a terrestrial laser scanner with analyses of georeferenced aerial photography for measuring erosion of bluffs in the Le Sueur watershed of Minnesota, USA. Eroding banks were digitized from aerial photographs for 243 bluffs, while laser scans were taken of 15 bluffs, and results were extrapolated to 480 bluffs. These bluffs were large enough to be identified using

3 m resolution elevation data and with a height up to 160 m. The study found an average erosion rate of 0.20 m year$^{-1}$ with the laser scanner and 0.14 m year$^{-1}$ from aerial photographs. An average difference of 36% was found between sediment loading measurements from the two techniques. Eltner et al. [24] compared a terrestrial laser scanner with unmanned aerial vehicle (UAV) photogrammetry for measuring bank erosion in two European catchments. They found that there was an average difference between point clouds of the laser scan and UAV photogrammetry of 3.1 to 18.0 mm, depending on the camera and software used for photogrammetry. Although we did not interpret aerial photography or include photogrammetry data in our analysis, the findings of Day and Eltner are relevant because they demonstrate the comparability of laser scanning with other techniques. Ours is the first study to compare erosion pins, a total station, and a terrestrial laser scanner on the same banks.

*1.3. Objectives*

Our objectives were to (1) evaluate and compare three techniques for quantifying streambank erosion: erosion pins, total station surveyor, and terrestrial laser scanning, (2) assess the spatial distribution of streambank erosion rates in the Indian Mill Creek watershed of Michigan, USA, and (3) estimate the annual rate of sediment loading in the watershed from streambank erosion and compare with modeled estimates from a nonpoint source pollution study. This research benefits watershed managers in addressing fish and macroinvertebrate community impairments in Indian Mill Creek and other watersheds that are degraded by excessive sediment. It also benefits owners of property and infrastructure along streambanks who can experience damages from streambank erosion.

## 2. Materials and Methods

*2.1. Study Area*

Indian Mill Creek in Kent County, Michigan, USA (HUC 040500060504), is a tributary to the Grand River and is 18.5 km long with a 44 km$^2$ watershed. The creek resides in the Southern Michigan Northern Indiana Till Plains ecoregion, characterized by irregular plains, cropland, pasture, and oak/hickory/beech/maple forests [25]. The watershed land cover is predominately urban (43%) and agricultural (39%), with commercial and residential development in the lower part, natural and urban lands in the middle, and farmland and orchards in the upper part [26]. This land cover pattern affects the distribution of erosion risk in the watershed. The National Weather Service classifies the area as a humid continental climate with distinct summers and winters and fairly even distribution of precipitation throughout the year (www.weather.gov, accessed 9 September, 2019). A total of 28.5 km of streams were identified in the watershed using a Geographic Information System (GIS) and United States Geological Survey 3DEP data (https://www.usgs.gov/core-science-systems/ngp/3dep, accessed 13 August, 2019).

*2.2. Site Design*

Nine sites were chosen for this study (Figure 1). Four sites were in the lower urbanized reaches of Indian Mill Creek, three sites were in the upper farmland, and two sites were on small urbanized tributaries. Sites are labeled with name and left (L) or right (R) bank and are ordered from lowest reach (IMC7) to headwaters (IMC1), followed by the small tributary sites (BC and WD). Sites were chosen to be dispersed around the watershed and where permission for access was obtained by landowners. For each site, an 18 m section of stream was chosen, based on a balance between an open channel for laser scanning and being representative of the reach, and then split into the left and right banks while looking downstream. Erosion pins were installed at all eighteen banks, total station surveys were performed at sixteen, and laser scanning was performed at ten. The presence/absence of undercut banks and heavy vegetation at each bank was also noted.

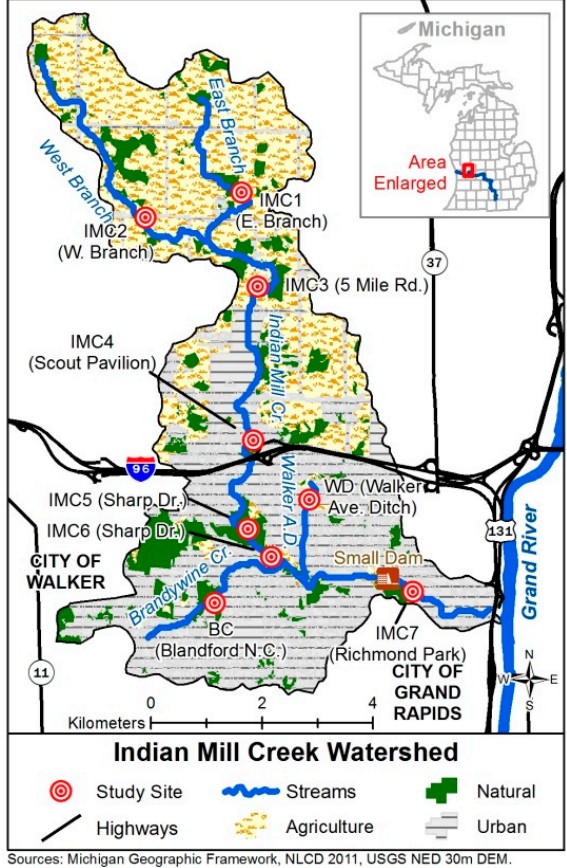

**Figure 1.** Study area map of the Indian Mill Creek watershed with features, land cover, and sites.

*2.3. Erosion Pins*

A total of 137 erosion pins were installed at the eighteen banks following the design of prior studies [5,11,27]. The erosion pins were of 2' × 0.5" rebar pieces. Prior to installing erosion pins, the 18 m stream section was divided into three six-meter subsections using a measuring tape. Erosion pins were carefully installed in the middle of each subsection on both banks. One to three pins were installed at each location evenly spaced up the bank, with one pin for approximately every meter of bank height (Figure 2). Extra pins were installed if there were visible changes not captured by the design, such as the vertical transition between an undercut bank and vegetated slope. The average number of erosion pins deployed along the longitudinal 18 m study streambanks was 7.3, with a minimum of 4 and maximum of 20 pins per bank. Thus, the erosion pins had the lowest resolution of the three techniques.

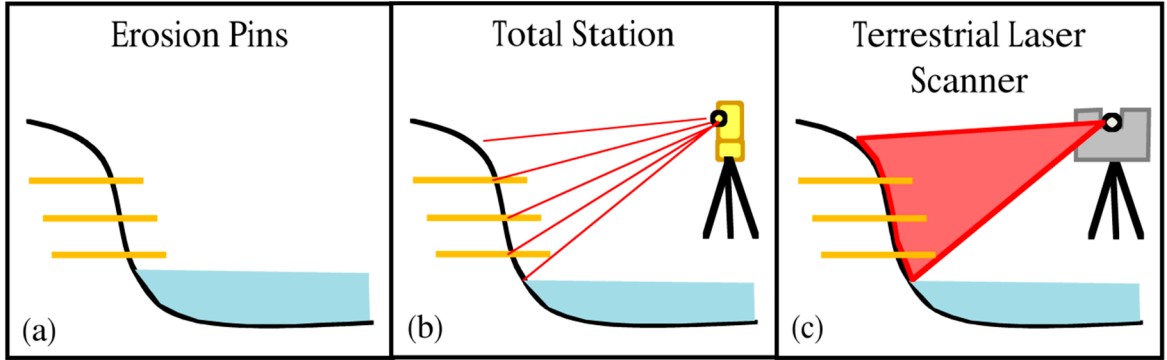

**Figure 2.** Deployment of (**a**) erosion pins, (**b**) total station, and (**c**) terrestrial laser scanner showing vertical profile of streambank.

Erosion pins were measured from tip of the pin to streambank using a measuring tape to the nearest 0.5 cm. The average of measurements from the top and bottom of the pin was used to account for bank slope. Where there was a longitudinal (horizontal) angle to the bank, the left and right sides of the pin were also included in the average. Erosion pins were measured monthly from May to September 2017, with two additional measurements following rain storms, then April to May 2018. The spread of erosion pin data at each site was analyzed using R3.3.2 [28]. The volume of soil loss per meter of stream length for each bank was estimated following methods of Palmer [29] and Zaimes et al. [30], by multiplying the average erosion pin value by the bank height, which we estimated as a uniform height from total station data. Overall change in bank volume was estimated by multiplying this rate by the 18 m site length.

### 2.4. Total Station

The first step of the total station surveys was to set four control points at each site using a Trimble Geo7x Global Positioning System (GPS) with Zephyr external antenna. These control points tie into the NAD 1983 UTM Zone 16 N projected coordinate system and orient the total station. Control points were two foot rebar stakes driven into the ground and marked with orange tape or a cap. Trimble TerraSync 5.86 software (Trimble Navigation Limited, Westminster, CO, USA) was used to collect data. All GPS data were post-processed in Trimble Pathfinder Office using data from the Grand Rapids Continuously Operating Reference Station, with estimated accuracies of <5 cm at most control points, but dropping to <30 cm at four wooded control points (one in RP, one in DU, and 2 in BC sites).

A Topcon GPT-3107 W total station theodolite on tripod with Spectra Precision SurveyPro software (Spectra Geospatial, Westminster, CO, USA) was used to survey streambanks in May 2017 and May 2018. The instrument was set on one of the control points and backsighted to the farthest point for the most accurate orientation. When the instrument needed to be moved, a temporary control point was created by pushing a marker into the ground, and the previous point was checkpointed to determine error during movement. A reflector prism was used on top of a staff with bubble level to collect points. Using this prism, the stated accuracy of the instrument was 3 mm. For undercut banks, the horizontal distance between the prism staff and the back of the undercut was noted. However, data for undercut banks were not incorporated into erosion estimates because virtual models could not account for overhanging bank shape.

The site design for the total station surveys was based on methods of Keim et al. [13] and Resop and Hession [9]. Seven transects were performed along each bank over the 18 m site, at the 0, 3, 6, 9, 12, 15, and 18 m marks. The 3, 9, and 15 m marks coincided with erosion pin locations. In each transect, sideshots for the top of the bank and toe were collected. Then, two to three shots were taken evenly spaced along the bank, depending on its size and variability (Figure 2). These shots were taken at erosion pins during the 3, 9, and 15 m transects, at the location where the pin met the streambank. The total station had an average of 35 measurement points per streambank (min = 23, max = 52), so collected higher resolution data than the erosion pins in this study.

Total station data were exported as a CSV file and imported into ESRI ArcMap software (Environmental Systems Research Institute, Redlands, CA, USA). Then, xy data were displayed and exported as a shapefile. A separate file was created for each streambank using the Select tool of ESRI ArcToolbox. A 3D TIN file was created using Create TIN with Delaunay Triangulation. The TINs were cropped using the Delineate TIN Data Area Tool of 3D Analyst to remove superfluous data. The volume of soil gain or loss between 2017 and 2018 TIN streambank models was then calculated using the Surface Difference Tool of ESRI 3D Analyst. The volume was divided by site length to estimate change in volume per meter of stream per year.

### 2.5. Terrestrial Laser Scanner

One to both banks were surveyed at each site with a FARO Focus3D terrestrial laser scanner in May 2017 and a Trimble TX8 scanner in May 2018 (Figure 2). The stated ranging accuracies of

the Focus3D and TX8 are under 2 mm. Ten total banks were chosen to incorporate representative conditions and have clear visibility. Three survey markers were placed along each bank, as far apart as possible without sacrificing visibility. Target spheres were placed on these markers with clear visibility to each scanning location. These markers act as control points, and were surveyed with the total station, so laser scan results can be projected in a Geographic Information System (GIS).

Next, a preliminary low-quality scan was taken to adjust the horizontal and vertical scan limits. Prior to the full scan, the resolution and quality were set. We used 1:1 resolution and 2× quality and color image for the FARO scans, and Level 3 quality for the Trimble scans. These levels were chosen because they were successfully used by the Annis Water Resources Institute previously (Kurt Thompson, personal communication) or recommended for the purposes of our study (Mark Tenhove, personal communication) as a balance between high quality data and manageable file size. Laser scans from both instruments were processed using CloudCompare software (http://www.cloudcompare.org, accessed 3 September, 2019), although Trimble scans first had to be exported to a compatible LAZ format using Trimble RealWorks 10.4.3 (Trimble Navigation Limited, Westminster, CO, USA). The FLS plugin was used to import FARO files to CloudCompare. Excess data were cut out and scans were aligned by target spheres. At the IMC7 and IMC1 sites, target markers disappeared so the alignment incorporated sturdy points on wood or metal structures, and manual alignment was needed for IMC7. The CANUPO plugin [22] and veget_LongRange.prm filter [31] with 0.1 m filtering resolution were then used to filter vegetation from the scans. This gave the most accurate classification of filters and filtering resolutions we experimented with and was within the processing capabilities of our computer. Other filters we experimented with were otira_vegetsuper.prm and otira_vegetsemi.prm [22], as well as vegetRangiCliff.prm and vegetTidal.prm [31].

Volume change of streambanks between 2017 and 2018 was calculated in Trimble RealWorks using the Volume Calculation tool with horizontal difference and 10 cm resolution. The percent of laser scan coverage from these volume outputs was calculated by dividing the scan area occupied by bank in both 2017 and 2018, facing the bank directly and horizontally from the stream, by the total gridded area of the file. The difference in laser scan coverage between banks with and without heavy vegetation was analyzed using a Shapiro–Wilk test to confirm normal distribution ($p = 0.110$ without vegetation, $p = 0.547$ with vegetation), followed by a t-test in R 3.3.2.

*2.6. Statistical Comparisons and Visualization*

Statistical tests were performed in R 3.3.2 using data from the ten banks that had laser scans. The IMC6 right bank was removed because it was deemed an outlier for the laser scan tests in terms of volumetric change, being 4.3 times higher than the second highest laser scan measurement, and affecting the normality. Shapiro–Wilk tests were used on the erosion pin, total station, and laser scanner volume change estimates to determine normality. Data from all three techniques were found to be normally distributed ($p = 0.977$, $0.964$, and $0.746$). Statistical differences between techniques were tested using Analysis of Variance (ANOVA) with randomized complete block design, with estimates of erosion rate as values, techniques as groups, and sites as blocks. Plots of normal Q-Q and residuals vs. fitted values were interpreted and suggested that the ANOVA was appropriate to use over data transformations or nonparametric alternatives. A similar ANOVA test was used by Purvis and Fox [32] to analyze the influence of riparian buffers and time period on erosion rates. Correlations between techniques were tested using Pearson Tests with Holm *p*-value adjustments for multiple comparisons. Relative error between volume results of the laser scanner and total station techniques were calculated to assess proportionately how close the results from the techniques typically were to each other, following the methods of Resop and Hession [9], who took the difference between laser scan and total station results, divided by the laser scan result. We calculated the relative error for laser scan and erosion pin results, and for erosion pin and total station results, in the same fashion. The IMC4 (L) bank was removed from the relative error analysis because it was an outlier with high total station error and less than 1% laser scan coverage after vegetation filtering.

### 2.7. Basin-Wide Estimates

Basin-wide estimates of sediment loading from bank erosion were derived by multiplying the bank erosion rate per meter of stream length ($m^3 \, m^{-1} \, year^{-1}$) by the entire length of streams in the watershed (28,500 m) by an average soil bulk density of eroding streambanks 1500 kg $(m^3)^{-1}$ [33]. We used erosion pin data to compare basin-wide estimates with other studies because the erosion pins had more sites and no limitations in coverage due to vegetation or other obstacles.

## 3. Results

### 3.1. Site Conditions, Erosion, and Deposition

Our study documented streambank conditions, volumetric changes using three erosion measurement techniques, and coverage of the laser scan data (Table 1). Negative bank volume change represents net erosion over the study period, while positive change represents net deposition. We found large absolute differences between the volume change measurements from the techniques. There was no discernible relationship between undercut banks and total station results biased toward deposition, possibly because the bias from undercut banks was relatively small compared to the spread of the data. The erosion pin, total station, and terrestrial laser scanner data used in these analyses are published through Mendeley Data at http://dx.doi.org/10.17632/th48ctg5ww.1 under a Creative Commons Attribution 4.0 International license.

**Table 1.** Site conditions, volumetric results, and laser scan coverage for study streambank in the Indian Mill Creek watershed.

| Location | Conditions | | Volume Change ($m^3 \, m^{-1} \, year^{-1}$) * | | | Coverage |
|---|---|---|---|---|---|---|
| Site (Bank) | Undercut Banks | Heavy Vegetation | Erosion Pins | Total Station | Laser Scanner | Laser (%) |
| IMC7 (L) | No | No | 0.081 | 0.264 | 0.015 | 21.4% |
| IMC7 (R) | No | No | 0.027 | 0.081 | 0.022 | 29.8% |
| IMC6 (L) | Yes | No | −0.004 | −0.065 | NA | NA |
| IMC6 (R) | Yes | No | −0.082 | −0.111 | 0.155 | 60.1% |
| IMC5 (L) | Yes | No | −0.105 | −0.078 | 0.004 | 24.4% |
| IMC5 (R) | Yes | No | −0.065 | 0.098 | 0.008 | 38.6% |
| IMC4 (L) | Yes | Yes | −0.034 | 0.047 | −0.001 | 0.5% |
| IMC4 (R) | No | No | 0.078 | 0.424 | NA | NA |
| IMC3 (L) | No | No | −0.070 | NA | NA | NA |
| IMC3 (R) | No | No | −0.048 | NA | NA | NA |
| IMC2 (L) | No | Yes | −0.003 | −0.018 | 0.001 | 5.6% |
| IMC2 (R) | Yes | No | −0.066 | −0.111 | NA | NA |
| IMC1 (L) | Yes | Yes | −0.034 | −0.055 | NA | NA |
| IMC1 (R) | Yes | Yes | −0.052 | −0.273 | −0.036 | 11.9% |
| WD (L) | No | Yes | 0.003 | 0.046 | NA | NA |
| WD (R) | Yes | Yes | −0.024 | −0.186 | −0.008 | 29.0% |
| BC (L) | No | No | −0.016 | 0.100 | NA | NA |
| BC (R) | Yes | No | −0.011 | 0.383 | 0.033 | 20.5% |

* NA's exist where a bank was not surveyed for logistical reasons.

### 3.2. Statistical Comparisons between Techniques

The ANOVA showed that there were no detectable differences between streambank erosion measurement techniques (df = 2/23, F = 0.457, $p$ = 0.639). Correlation tests found no significant correlations between erosion pin and total station data (R = 0.51, $p$ = 0.330) or erosion pin and laser scanner data (R = 0.40, $p$ = 0.330; Figure 3). However, there was a significant correlation between total station and laser scan data (R = 0.89, $p$ = 0.003). These correlations show how measurements are related, but do not indicate which is more accurate.

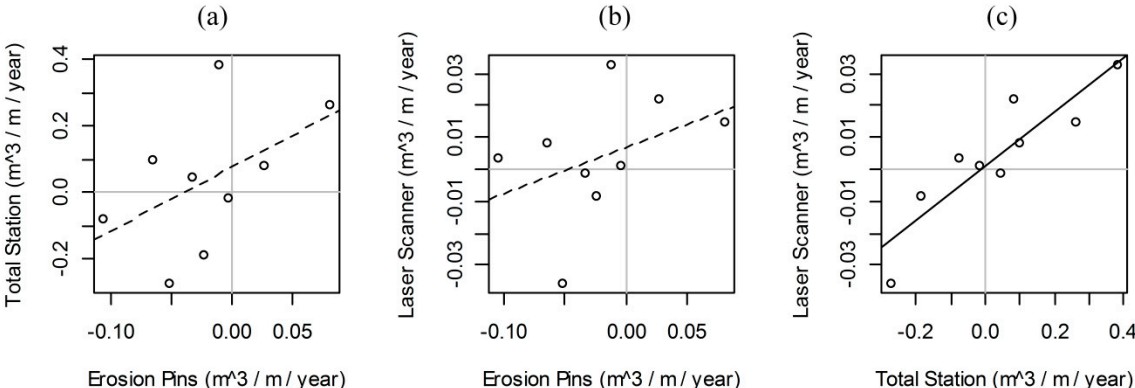

**Figure 3.** Correlations of bank volume change rate estimates between (**a**) erosion pins and total station (R = 0.51, *p* = 0.330), (**b**) erosion pins and laser scanner (R = 0.40, *p* = 0.330), and (**c**) total station and laser scanner (R = 0.89, *p* = 0.003) for nine sites in the Indian Mill Creek watershed. Solid line indicates significant correlation.

### 3.3. Vegetation Filtering

The terrestrial laser scanner performed well on barren streambanks with clear line of sight. It collected high resolution, quality data for these banks (see Section 3.8 for estimates of error). After processing, the 2017 scans had a mean of 4,482,985 data points representing streambank (min = 909,411, max = 27,449,058), while the 2018 scans had a mean of 1,564,049 points (min = 148,535, max = 3,775,508). Thus, the laser scanner had much higher resolution than the erosion pins or total station for barren banks, but was strongly affected by data gaps where the streambank was obscured by vegetation or other obstructions. Banks with heavy vegetation had significantly lower average laser scan coverage after vegetation filtering (11.75%) than other banks (32.5%, *p* = 0.047) (Table 1). These banks were most common in agricultural headwaters, which had substantial herbaceous plant growth, especially in the spring. Laser scanner data could be underestimating change in bank volume because erosion of banks behind vegetation, roots, and other obstructions was not accounted for. This was especially true at the IMC4 (L) bank (Figure 4g), where only 0.5% of the bank had coverage. This site was characterized by large masses of roots and overhanging vegetation that obscured the bank and were removed by the vegetation filter. The ability for the laser scanner to produce high coverage along vegetated streambanks was a significant limitation of the technique. As far as we know, there is no standard for when coverage becomes too small to reliably use laser scan data. The site with the highest percent laser coverage, IMC6 (R), was a steep bank under forest canopy that was mostly clear of small vegetation growth and other obstructions. The IMC7 (R) bank was assigned a classification of no heavy vegetation because open banks were observed; however, shrubs and exposed roots could be responsible for low laser scan coverage.

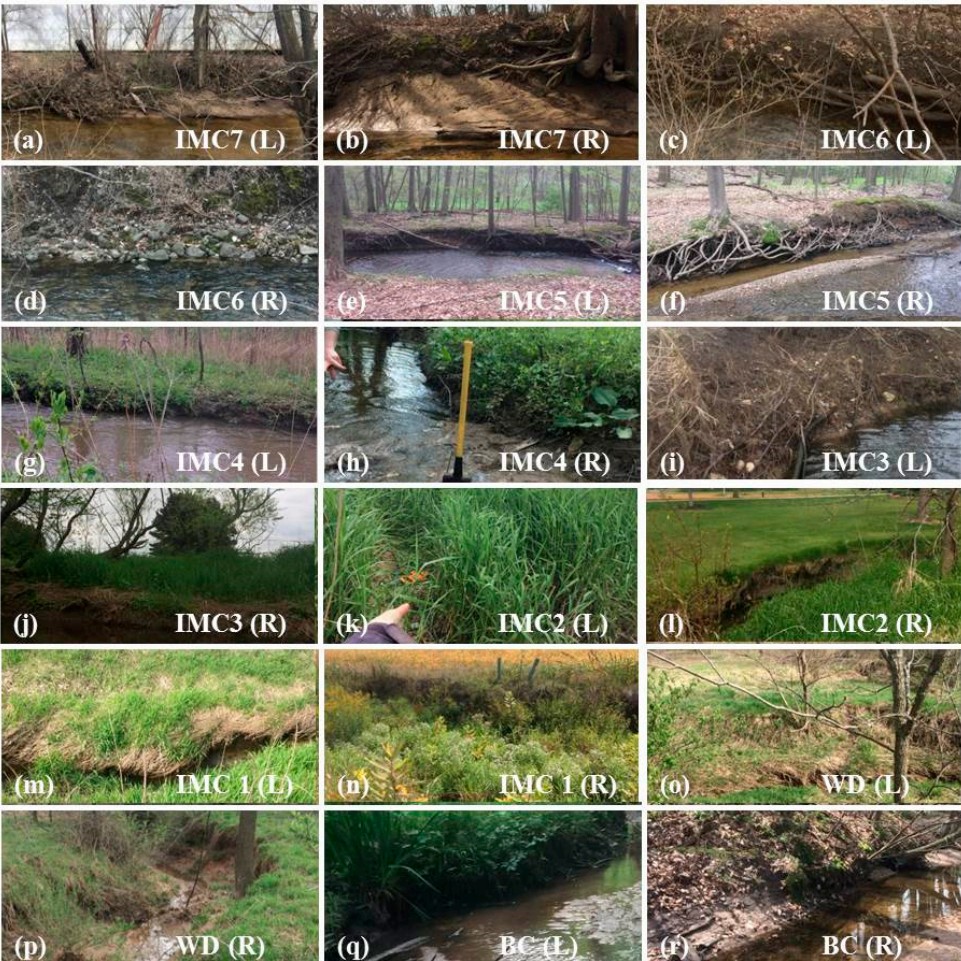

**Figure 4.** Photos of the 18 study streambanks in the Indian Mill Creek watershed, labeled by figure letter, site and left (L) or right (R) bank. Photos (**a**) through (**h**) are in the lower watershed through urban and forested land cover, (**i**) through (**n**) are in the upper watershed through farmland, and (**o**) through (**r**) are on tributaries.

*3.4. Comparative Analyses of Techniques and Sites*

Study streambanks experienced net deposition (positive volume change), net erosion (negative), little change in bank volume (points near zero), or a mixture depending on the technique (Figure 5). Relative error between techniques were substantial, with an average difference of 650% between erosion pins and total station data, 596% between the laser scanner and erosion pins, and 1275% between the laser scanner and total station (Table 2).

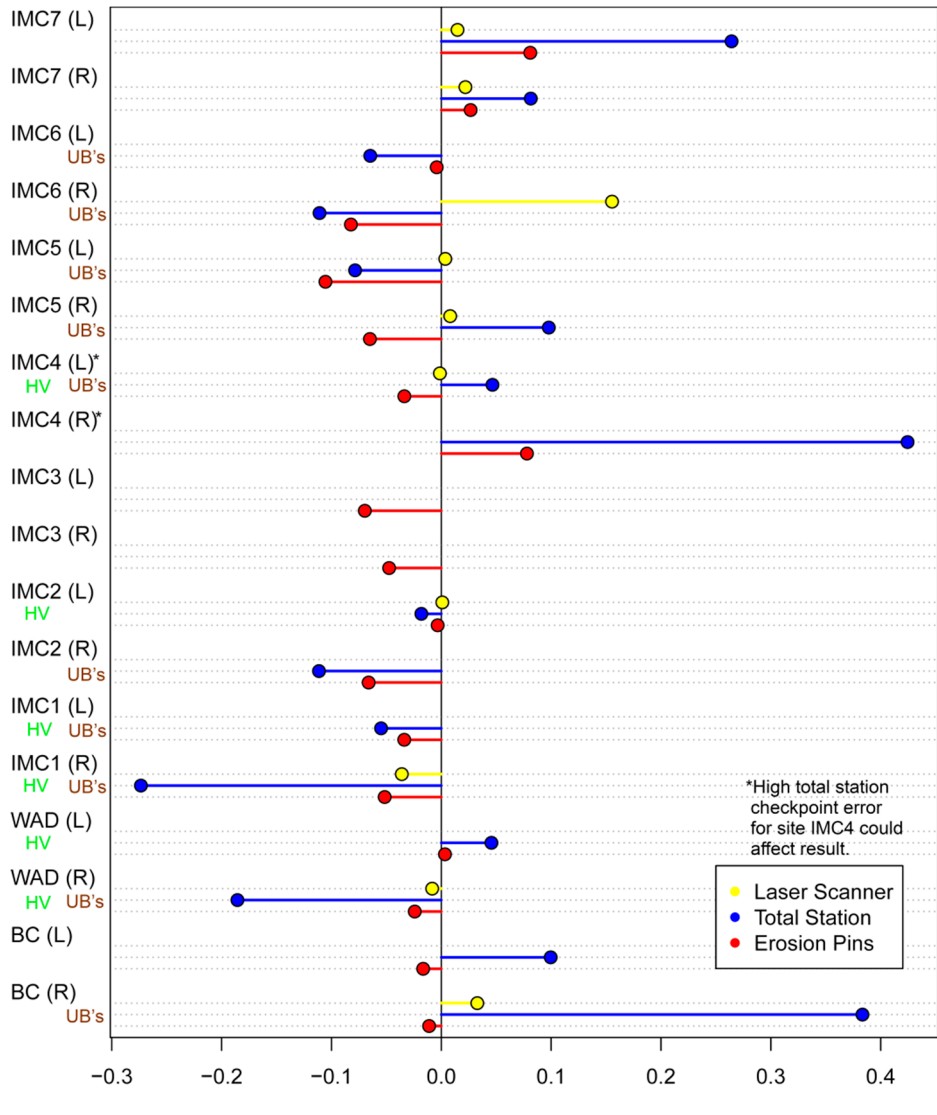

**Figure 5.** Comparison of results from techniques used to measure streambank erosion in the Indian Mill Creek watershed 2017–2018. Positive values indicate net deposition while negative values indicate net erosion being measured. Presence of heavy vegetation (HV) or undercut banks (UB's) is noted under site names to visualize the effects of these variables on estimates of bank volume change.

**Table 2.** Relative error in volume results for techniques to measure streambank change in the Indian Mill Creek watershed, calculated only for sites that had all three techniques used. Reference Table 1 for absolute values.

| Site (Bank) | Erosion Pins and Total Station | Laser Scanner and Erosion Pins | Laser Scanner and Total Station |
|:-:|:-:|:-:|:-:|
| IMC7 (L) | 226% | 449% | 1692% |
| IMC7 (R) | 205% | 22% | 271% |
| IMC6 (R) | 35% | 153% | 171% |
| IMC5 (L) | 26% | 3003% | 2260% |
| IMC5 (R) | 251% | 904% | 1111% |
| IMC4 (L) | 238% | 2511% | 3715% |
| IMC2 (L) | 448% | 466% | 2106% |
| IMC1 (R) | 430% | 43% | 661% |
| WD (R) | 668% | 191% | 2136% |
| BC (R) | 3559% | 134% | 1070% |

### 3.5. Lower Watershed Sites (IMC7, IMC6, IMC5, and IMC4)

Sites in the lower watershed experienced either a positive volume change or negative depending on the bank and technique. Erosion pins, total station, and laser scanner all documented deposition of sediment at both IMC7 banks (Figure 4a,b), although there was considerable relative error between rates. Erosion pin and total station results were similar for the IMC6 (L) site, showing only slight bank erosion (Figure 4c. At the IMC6 (R) site, the laser scanner measured high deposition of sediment on the bank, while the total station and erosion pins both measured substantial erosion (Figure 4d). The relative errors between techniques at this site were under 200%, however, because measurements were all of roughly the same magnitude. The difference in measurements here could be because erosional areas were shadowed by leafy shrubs.

At both IMC5 banks, the laser scanner documented very little change in bank volume, even though there was substantial undercutting and slumping along both banks (Figure 4e,f), documented by erosion pins. The total station estimate was consistent with erosion pin data (26% difference). However, for the IMC5 (R) bank, there was a 251% difference. A likely reason for the disparity is that the entire right bank was undercut and the lip had been pushed up; erosion pins were still able to collect data in the undercut, but the total station with was only able to collect data on the top of the bank.

At the IMC4 (L) bank, erosion pins measured slight erosion, while the total station estimated slight deposition. This difference could once again be the undercuts that extend the entire length (Figure 4g). Reduced laser scan coverage from roots and vegetation (0.5%) likely explains the low estimate of bank change from the laser scanner. The IMC4 (R) bank had a disparity where the total station measured heavy sediment deposition, but the erosion pins only measured slight deposition. This could be because erosion pins are limited in their ability to measure localized sediment deposition (Figure 4h).

### 3.6. Upper Watershed Sites (IMC3, IMC2, and IMC1)

Agricultural sites in the upper watershed primarily experienced bank erosion. The IMC3 site had substantial bank erosion measured by erosion pins along both banks (Figure 4i,j). At this site, a constricting culvert under a driveway, large willow fallen across the creek, and runoff from upstream farmland could be altering the local hydrology to scour the banks. At the IMC2 site, the left bank had consistent measurements of bank change, showing slight erosion, although it is likely that the low estimates inflated the relative error between techniques (Figure 4k). This low erosion rate made sense because the site has a vegetated riparian buffer of approximately ten meters to protect the banks. Erosion pin and total station estimates for the IMC2 (R) bank both showed erosion. This bank was along a lawn with no riparian buffer and was visibly eroding (Figure 4l).

The IMC1 site also had visible erosion that was documented by all three techniques at the right bank, and both techniques used at the left. The total station estimated a much higher erosion rate at the right bank than the laser scan and erosion pins (relative errors of 661% and 430%), which could be because of the resolution and coverage of the data. This bank was heavily vegetated and had low laser scan coverage. Differences could also be due to bank shape, which was complex with many bends, slumps, and large barren areas (Figure 4m,n). Differences could also be caused by a high checkpoint error of the total station, possibly due to unstable soil conditions for the tripod.

### 3.7. Tributary Sites (WD and BC)

Small tributaries had a mixture of erosion and deposition. The WD site was along a meander, which explains why the left bank inside the bend had measured deposition, while the right bank on the outside of the bend had measured erosion (Figure 4o,p). The reason that the total station estimated more erosion for the WD (R) bank than the pins and laser scans (relative errors of 668% and 2136%) was because the laser scans had data gaps due to shrubs and herbaceous vegetation, and because the

erosion pins had coarser data that could miss eroding areas. At the BC site banks, the erosion pins measured erosion, while to total station measured deposition. The laser scanner data on the right bank also showed deposition, but of lesser magnitude. We observed sediment deposition on the bed of Brandywine Creek and the toe of the banks at the BC site, as well as evidence of powerful flows during storms that pushed down grass in the creek's floodplain. Undercut banks could also explain why the total station estimated more deposition (Figure 4q,r).

### 3.8. Estimates of Error

Total station end checkpoint error data show that measurements can vary by millimeters or centimeters (Table 3), with an average error of 5.5 cm (standard deviation 11.7 cm). The high 2018 checkpoint elevation error introduces uncertainty into the total station results for the IMC4 site. We presume that this error occurred because the tripod was set in soft muddy soil, causing the instrument to tilt during the survey. It could also have been a recording error because both the northing and easting error were small. Laser scanner alignment had an average of 0.7 cm error (standard deviation = 0.4 cm).

**Table 3.** End checkpoint error data from the total station surveys showing how much the instrument erred between the beginning and end of a streambank survey, along with alignment error from laser scanner targets in the Indian Mill Creek watershed 2017–2018. The IMC4 site was measured with only erosion pins so is not included.

| Location | Checkpoint Error 2017 (m) | | | Checkpoint Error 2018 (m) | | | Target Alignment Error (m) | | |
|---|---|---|---|---|---|---|---|---|---|
| Site | Northing | Easting | Elevation | Northing | Easting | Elevation | 1 | 2 | 3 |
| IMC7 | 0.008 | 0.006 | 0.064 | 0.009 | 0.011 | −0.006 | 0.008 | 0.010 | 0.002 |
| IMC6 | 0.004 | 0.001 | −0.004 | 0.392 | −0.016 | −0.005 | 0.006 | 0.004 | 0.009 |
| IMC5 | No data | No data | No data | −0.242 | −0.356 | −0.008 | 0.005 | 0.009 | 0.002 |
| IMC4 | −0.012 | 0.021 | −0.001 | 0.006 | 0.038 | −0.577 | 0.005 | 0.011 | 0.006 |
| IMC2 | −0.002 | −0.012 | −0.007 | −0.023 | −0.023 | −0.021 | 0.002 | 0.002 | 0.002 |
| IMC1 | 0.004 | −0.001 | 0.004 | 0.020 | −0.050 | 0.084 | 0.013 | 0.004 | 0.011 |
| WD | 0.022 | 0.022 | −0.014 | −0.034 | −0.012 | 0.024 | 0.014 | 0.012 | 0.011 |
| BC | No data | No data | No data | 0.054 | −0.018 | −0.051 | 0.011 | 0.011 | 0.011 |

### 3.9. Basin-Wide Estimates

Overall, an average bank volume change rate of −0.024 $m^3$ $m^{-1}$ $year^{-1}$ (standard deviation 0.049) was estimated from erosion pin data (Table 1). Both the total station and the laser scanner showed more deposition of sediment on streambanks, with an average bank volume change of 0.034 and 0.019 $m^3$ $m^{-1}$ $year^{-1}$, and standard deviation of 0.187 and 0.049. The high standard deviation and bank change rate from total station data is due in part to the right bank of the IMC4 site (Figure 4h). This bank is the inside of a meander bend with heavy deposition of sediment visible. This deposition was also documented with erosion pin data. It is possible that deposition of sediment on most banks from laser scanner data was due to vegetation and other obstructions shadowing eroding areas.

Assuming that the average erosion rate of our eighteen study banks from erosion pin data (0.024 $m^3$ $m^{-1}$ $year^{-1}$) represents the average bank erosion rate for the 28.5 km of streams of the Indian Mill Creek watershed, we estimate from erosion pin data that bank erosion contributes 1346.5 cubic meters of sediment per year to Indian Mill Creek. Multiplying by an average soil bulk density of eroded sediment of 1500 kg $(m^3)^{-1}$ [33], we estimate that streambank erosion contributes an annual load of 2020 Mg of sediment per year to Indian Mill Creek.

## 4. Discussion

### 4.1. Comparison of Techniques

We evaluated and compared three techniques for measuring streambank erosion: erosion pins, total station, and terrestrial laser scanner. We were unable to detect significant differences between measurement techniques and found a significant correlation only between total station and laser scanner data, suggesting that a larger change detected with the laser scanner would likely be larger with the total station as well. However, a small sample size could be a limitation of these comparisons, as it would reduce our power to find significant differences between techniques. Also, although the differences between techniques were not significant and one correlation was found, absolute and relative errors between techniques were substantial and could lead to meaningfully different conclusions about bank erosion. Thus, when designing a streambank erosion study, results between different techniques could have limited comparability, and thoughtful selection of methodology is very important depending on riparian conditions.

Our results show that selection of a streambank erosion measuring technique should be dependent on the goals of the project, resources available, desired resolution of data, and site conditions. Terrestrial laser scanning has high resolution and can detect small erosion rates with sub-centimeter error on open streambanks with little vegetation. The scanner itself is easy to use, requiring little effort for a high-resolution scan. However, the cost of the laser scanner would make it unusable for many watershed studies. The reason that laser scans were performed at fewer banks is that we were limited by time and financial resources to scan ten banks, while we had greater flexibility with erosion pins and the total station coverage. Additionally, training with special point cloud processing software, and ideally Geographic Information Systems, is necessary to process the laser scanner data. The terrestrial laser scanner performed well on clear barren streambanks, such as the right bank of IMC6. However, there were large data gaps and limited coverage when vegetation or other obstructions obscured the bank. This introduces uncertainty into the estimates of bank erosion because it is unclear how the bank is changing behind the vegetation. We recommend using the laser scanner only for bare banks with limited vegetation cover. If vegetated banks must be scanned, we recommend scanning them in early spring directly after snowmelt before vegetation has become established. We do not recommend physically removing vegetation from the banks because this could affect bank stability.

The total station or erosion pins are preferable techniques for vegetated banks. The pointed staff and reflector of the total station allowed us to collect data for points obstructed by vegetation. Similarly, erosion pins can be installed and measured on vegetated banks without loss of data. In general, erosion pins are the most cost-effective technique to measure streambank erosion. They can be installed and monitored for $1–2 per pin and do not require expensive equipment or familiarity of special software. However, they provide very low spatial resolution; our transects were spaced three meters apart with approximately one pin per meter bank height. We also observed minor destabilization of the bank while installing and checking the pins. The total station works effectively for barren or vegetated streambanks, however, it requires surveying skills, familiarity with the instrument and software, and may not readily be available to researchers. Additionally, minor bank destabilization can occur when using the staff and prism to collect data.

The total station does not work for undercut banks using the methods we performed, ignoring the space under the overhang in its entirety. Undercut banks were documented at the IMC6, IMC5, IMC4, IMC2, IMC1, WD, and BC sites. Although it is unclear how strongly they affected erosion estimates, these undercuts shifted total station data at these sites toward deposition because the undercutting erosion was ignored in the TIN model. Total station results also had a larger spread of data than the other techniques. While results from the laser scanner and erosion pins tended to show change less than 0.1 $m^3$ $m^{-1}$ $year^{-1}$, the total station results were more variable, estimating changes in bank volume up to 0.2 to 0.4 $m^3$ $m^{-1}$ $year^{-1}$ (Table 1, Figure 5). The BC site right bank, IMC7 left bank, and IMC4 right bank all had high deposition documented with a total station that was not consistent

with laser scanner or erosion pin results. The lack of erosion measurements from undercut banks could contribute to these high deposition estimates. On the other hand, the IMC1 right bank and WD right bank had relatively high erosion rates from total station data. An explanation for these rates could be from heavily eroding banks that were measured with the total station, but could have been between erosion pin transects or hidden from the laser scanner behind vegetation.

Resop and Hession [9] noted that measurement of bank erosion can involve large errors and uncertainty. They did not find any systematic differences between results of total station surveys and laser scans, aside from some instances where the total station could not collect data beneath an undercut bank. Our study supports this, as we found large absolute and relative errors between the techniques, but the ANOVA was unable to detect significant differences to suggest these errors were not due to chance. Resop and Hession found that volumes of soil erosion from their study streambank estimated by the total station and laser scanner had an average difference of 109%, with a range from 7% to 373%. That is much smaller than what we experienced between the laser scanner and total station, which had an average difference of 1275% with a range from 171% to 2260%. Vegetation and other complexities along our banks are likely responsible for this greater range of differences; the bank that Resop and Hession studied was bare, with little vegetation.

Potentially, multiple techniques could be used in a streambank erosion study to provide a more comprehensive analysis. Erosion pins could be used extensively in a basin to obtain measurements of streambank erosion at banks with varying conditions (e.g., vegetation, undercutting, or barren), focusing on representative measurements across the basin. The erosion pin data could also be used to screen for stream reaches that need more management or control. The terrestrial laser scanner could then be used to collect detailed site-specific measurements at these priority eroding banks, especially if they are barren and clear of obstructions. This would target financial resources and experienced staff to the sites that are most in need of erosion control. The combination of erosion pins and a terrestrial laser scanner in a focused study design could help reduce the complexity and challenges of assessing streambank erosion in a basin.

### 4.2. Spatial Distribution of Bank Erosion

We assessed the spatial distribution of streambank erosion in the Indian Mill Creek watershed. The lower watershed experienced net deposition of sediment along the banks (Figure 6), as we noted while observing heavy sand deposition on the IMC7 banks. This 5 km section of the creek (up from the mouth) has a low average gradient of approximately 0.3%, which could explain why the stream channel is aggrading there as the creek loses power [34,35]. This could also be an impact of the urbanization in the watershed, which can increase hillslope erosion upstream and cause the stream channel to aggrade [2]. Two sites (IMC4 and WD) experienced alternating patterns of erosion and deposition at opposite streambanks in meander segments. Fluvial processes along bends caused the outer bank to erode while sediment was deposited along the inner bank [36]. Stream processes have been identified as the dominant mechanisms affecting streambank erosion rates in a temperate watershed, more so than land use, subaerial (e.g., freeze/thaw) processes, or longitudinal location in a stream network [37]. All other sites experienced net bank erosion and contributed to sediment loading in the Indian Mill Creek watershed. Erosion in the agricultural headwaters of the creek (IMC1 and IMC2) could be intensified by riparian vegetation clearing and hydrological alterations [1]. The highest rates of bank erosion from erosion pin data were at the IMC6 and IMC5 sites, which are along a higher (0.7%) gradient section of the creek as it descends the Grand River valley. The higher gradient increases the power of the creek to degrade the stream channel [34,35].

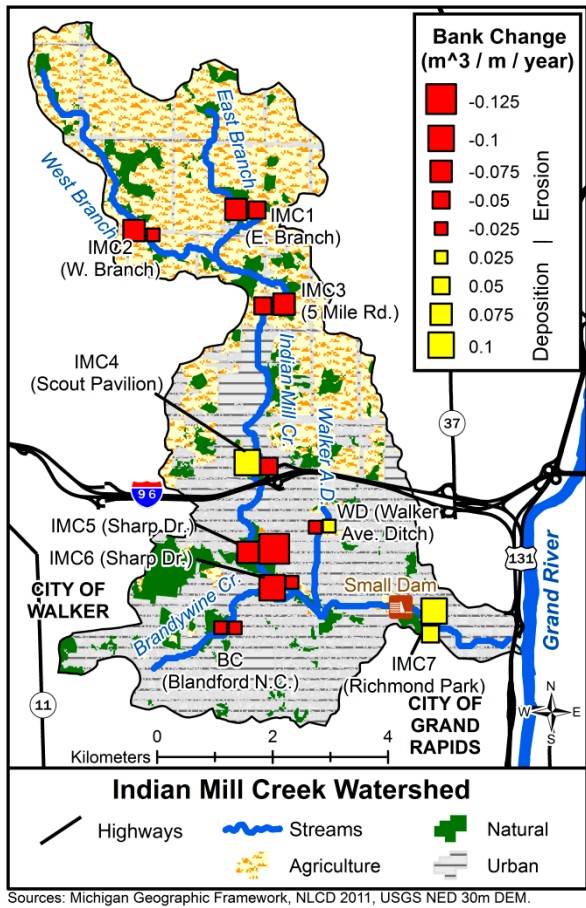

**Figure 6.** Spatial distribution of erosion (red) and deposition (yellow) rates for study streambanks using erosion pin results.

Understanding fluvial processes is important for studies investigating sediment loading sources such as ours [4]. Thus, one challenge for researchers who are studying sediment loading from streambank erosion at the watershed scale is to ensure that the action of fluvial processes and monitoring sites are representative of watershed conditions. Consequently, Kessler et al. [38] recommend that extrapolation of discrete streambank erosion measurements to estimate sediment loading in a river system should be avoided. However, the techniques used by the Kessler study (historic orthoimagery and plat maps) were different from ours, particularly in that they focus on relative soil loss from streambanks over time, which (although meeting the objectives of their study) is different from our determination of absolute soil losses at sites dispersed throughout the watershed.

*4.3. Estimation of Sediment Loading*

Both the total station and terrestrial laser scanner estimated an average bank volume change in the watershed that was positive, suggesting that more sediment was deposited on streambanks than was removed by streambank erosion, which seems unlikely and could be an effect of the uncertainties and limitations of the techniques, such as vegetation coverage and other obstructions for the laser scanner or undercutting for the total station. Additionally, the errors we found within techniques (e.g., checkpoint error or target alignment, Table 3) were much less than the errors between techniques (Figure 5), which supports using only one technique for making basin-wide estimates. We estimated the total load of sediment entering Indian Mill Creek from streambank erosion using erosion pin data (Section 2.7) and compared it to results from a concurrent study of field and streambank erosion rates that we have performed in the watershed. This concurrent study used the Enhanced Generalized Watershed Loading Functions (GWLF-E) model [33] for the time period 1997–2010. The model predicted that

average annual sediment loading from streambank erosion in the watershed during that time period was 1031.3 Mg year$^{-1}$, while annual sediment loading from field erosion was 5077.9 Mg year$^{-1}$. Our estimate of the contribution of sediment loading to Indian Mill Creek from the erosion pin data was 2020 Mg year$^{-1}$. This is roughly double the streambank erosion predictions of the GWLF-E model. The difference between our estimate and modeled predictions could be because the GWLF-E model was validated by watersheds in Pennsylvania that could have different conditions than Indian Mill Creek, such as topography, soils, and land cover. Stream discharge data collected with a flow meter suggest that GWLF-E, although not calibrated to Indian Mill Creek, follows the same pattern of increasing discharge toward its outlet, but may be overestimating discharge in subbasins by a factor of 2.8 to 11.0. The difference could also be that our eighteen study banks sample only a small proportion of the overall length of bank in Indian Mill Creek. Our best estimate of sediment loading from bank erosion in relation to the GWLF-E field erosion estimate suggests that streambank erosion contributes 28.5% of the annual total sediment load to Indian Mill Creek. This is a substantial portion of the sediment load and is almost certainly affecting the quality of aquatic habitat, fish, and macroinvertebrate communities in the Indian Mill Creek watershed, which was identified by the Michigan Department of Environment, Great Lakes, and Energy to be impaired with degraded fish and macroinvertebrate communities, with sediment problems as the cause [26,39].

Previous studies have demonstrated that streambank erosion can be a large source of sediment loading in a watershed [40]. Kiesel et al. [5] estimated for a lowland catchment in Germany that 71% of the sediment load was from streambank erosion. The catchment was relatively flat but had a large amount of agriculture along the creek. Kiesel found this estimate to be plausible because it was similar to estimates for other European catchments. Evans et al. [33] modeled the contribution of streambank erosion to 28 Pennsylvania, USA watersheds, including watersheds with extensive agricultural and urban land cover, using the GWLF-E model and estimated that eroding banks contributed between 4.8% and 78.6% of the total sediment loads to those watersheds, with an average of 17.9%. Fox et al. [4] reviewed fourteen studies and found that bank erosion contributions ranged from 7% to 92% of the suspended sediment load in the study watersheds. Beck et al. [41] estimated that bank erosion contributes 4% to 44% of annual suspended sediment load in an agricultural Iowa, USA, watershed. Our estimate that 28.5% of the total sediment load in Indian Mill Creek comes from eroding banks is reasonable compared with these studies because it is well within the published ranges, in particular for the agricultural and urban watersheds in the nearby states of Iowa and Pennsylvania.

## 5. Conclusions

Sediment pollution is a major concern for streams throughout the United States. One difficulty in managing sediment pollution in streams is that it is difficult to quantify sediment from streambank erosion. We evaluated three techniques for measuring streambank erosion at nine sites in the Indian Mill Creek watershed: erosion pins, total station surveyor, and terrestrial laser scanner. Although we found substantial absolute and relative errors in comparisons between measurement techniques, we were unable to detect significant differences, and found a significant correlation only between total station and laser scanner data. This suggests that the techniques may have limited comparability. Each technique had advantages and disadvantages for measuring eroding streambanks, suggesting that their application is highly dependent on watershed and site-specific conditions. Erosion pins and total station surveying can be used in vegetated banks but have coarse resolution, while laser scanning has high resolution but cannot measure through dense streambank vegetation even when employing the vegetation filter. Ultimately, the choice of technique depends on the goals of the project, bank conditions, desired resolution, and the resources available. We also assessed how streambank erosion rates varied spatially throughout the watershed, with the most deposition occurring in the lowest two kilometers of Indian Mill Creek, and the most erosion in middle to upper reaches. Overall, we estimate that streambank erosion contributed 2020 Mg of sediment each year to Indian Mill Creek, which is 28.5% of modelled sediment loads. This estimate shows that bank erosion is a substantial

portion of the total sediment load and is almost certainly affecting the quality of aquatic habitat, fish, and macroinvertebrate communities in the Indian Mill Creek watershed.

**Author Contributions:** Conceptualization, D.T.M., R.R.R. and J.N.M.; Data curation, D.T.M. and J.N.M.; Formal analysis, D.T.M., R.R.R. and J.N.M.; Funding acquisition, D.T.M., R.R.R. and J.N.M.; Investigation, D.T.M. and R.R.R.; Methodology, D.T.M., R.R.R. and J.N.M.; Project administration, D.T.M., R.R.R. and J.N.M.; Resources, D.T.M., R.R.R. and J.N.M.; Software, D.T.M. and J.N.M.; Supervision, R.R.R. and J.N.M.; Validation, D.T.M., R.R.R. and J.N.M.; Visualization, D.T.M., R.R.R. and J.N.M.; Writing—original draft, D.T.M., R.R.R. and J.N.M.; Writing—review & editing, D.T.M., R.R.R. and J.N.M.

**Funding:** This work was supported by the Grand Valley State University Graduate School, Annis Water Resources Institute, and Lower Grand River Organization of Watersheds.

**Acknowledgments:** We would like to thank Kurt Thompson, Peter Wampler, Wendy Ogilvie, Michigan Surveyors Supply Company, Matt Allen, Noah Cleghorn, John Koches, Dana Strouse, Molly Lane, Justin VanPaemel, Rachel Frantz, Eileen Boekestein, Jacob Gardner, and Rajesh Sigdel for assistance.

**Conflicts of Interest:** The authors declare no conflict of interest.

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
