# Peer review of "Measuring Streambank Erosion: A Comparison of Erosion Pins, Total Station, and Terrestrial Laser Scanner"

_water, doi:10.3390/w11091846_

Round 1

Reviewer 1 Report

This manuscript presents a comparison of three techniques for measuring streambank erosion. The manuscript is very well structured and straight to the point. This research is highly interesting, thus I recommend this manuscript for publication in Water. However, I suggest some minor revisions below:

Section 2.2: How was the selection of the nine sites done? Randomly?

Sections 2.3-2.5: Please provide the manufacturer names of the softwares and instruments used in this study. Also, some figures with the field deployment of the different techniques would be helpful for the reader. What is the accuracy and resolution of the instruments used?

Section 3.2: Is the small sample size a limitation of the statistical comparison? If yes, it should be stated as a potential limitation of this analysis

Author Response

Reviewer 1 Responses (water-571000):

This manuscript presents a comparison of three techniques for measuring streambank erosion. The manuscript is very well structured and straight to the point. This research is highly interesting, thus I recommend this manuscript for publication in Water. However, I suggest some minor revisions below:  Response: We thank the reviewer for the helpful improvements to the manuscript. We have made the suggested revisions, which have added considerably to the quality of the manuscript. We have noted changes in red in the revised manuscript file showing edits. Section 2.2: How was the selection of the nine sites done? Randomly? Response: In our revised manuscript, we have added text to Section 2.2: “Sites were chosen to be dispersed around the watershed and where permission for access was obtained by landowners.” Sections 2.3-2.5: Please provide the manufacturer names of the softwares and instruments used in this study. Also, some figures with the field deployment of the different techniques would be helpful for the reader. What is the accuracy and resolution of the instruments used? Response: To provide more details about manufacturer names and software, we added text to Section 2.4: “Trimble TerraSync 5.86,” “Trimble Pathfinder Office,” “Spectra Precision SurveyPro,” “ESRI ArcMap,” “ESRI ArcToolbox,” “ESRI 3D Analyst,” and “CloudCompare software (http://www.cloudcompare.org).” We also added text to Section 2.3: “The erosion pins were 2’ x 0.5” rebar pieces.” We produced a figure (Figure 2) that shows the field deployment of our techniques along a vertical profile of streambank. We appreciate this suggestion. To better describe the accuracy and of the measurements, we also added text to: The first paragraph of Section 2.4 detailing the accuracies of our GPS coordinates of control points: “…, with estimated accuracies of <5 cm at most control points, but dropping to <30 cm at four wooded control points (one in RP, one in DU, and 2 in BC sites).” The second paragraph of Section 2.4 detailing the stated accuracy of the Topcon total station: “Using this prism, the stated accuracy of the instrument was 3 mm.” The first paragraph of 2.5 detailing the stated the laser scanner accuracies: “The stated ranging accuracies of the Focus3D and TX8 are under 2mm.” To compare the resolution of the three measurement techniques, we described the numbers of points along the study streambanks: We added text to the first paragraph of Section 2.3 detailing the deployment of erosion pins along each streambank: “The average number of erosion pins deployed along the longitudinal 18 m study streambanks was 7.3, with a minimum of 4 and maximum of 20 pins per bank. Thus, the erosion pins had the lowest resolution of the three techniques.” We added a statement to Section 2.4, third paragraph: “The total station had an average of 35 measurement points per streambank (min = 23, max = 52), so collected higher resolution data than the erosion pins in this study.” We added a statement to the first paragraph of Section 3.3: “After processing, the 2017 scans had a mean of 7,482,985 data points representing streambank (min = 909,411, max = 27,449,058), while the 2018 scans had a mean of 1,564,049 points (min = 148,535, max = 3,775,508). Thus, the laser scanner had much higher resolution than the erosion pins or total station for barren banks, but was affected by data gaps where the streambank was obscured by vegetation or other obstructions.” Section 3.2: Is the small sample size a limitation of the statistical comparison? If yes, it should be stated as a potential limitation of this analysis Response: Thank you for suggesting this improvement. We added text to Section 4.1: “However, a small sample size could be a limitation of this comparison.”

Reviewer 2 Report

General Comments for the Authors:

This paper compares different streambank erosion measurement techniques in an effort to highlight the utility of different measurement methodologies (erosion pins, total station, and TLS) across a range of bank conditions. The authors seek to not only compare the data gathered from each method but to also evaluate each method in terms of how well it represents the streambank changes over time. The authors then use their erosion measurements to derive a catchment sediment load attributable to streambank erosion.

The fundamental field surveys and laboratory analyses are sound; however, there are several issues with how the results are presented and discussed. Additionally, the writing of this manuscript, while mostly grammatically correct, contains numerous redundancies, overly verbose, and unclear phrasing that detracts from the message that the authors seek to convey. A few general issues are listed below, and I refer you to the manuscript file for in-text comments.

1. The language describing the ANOVA and individual correlation tests is unclear. The authors use the word ‘differences’ in place of ‘correlation’, and they use ‘correlation’ to indicate similarity in the data. The axes of the graphs in Figure 2 demonstrate this point. A statistically significant correlation was found between the total station and TLS data (Figure 2 c). However, the data from these two measurement methods are an order of magnitude off, as seen in the range of the axes. This correlation is only valuable if you are trying to predict the results of one method by employing the other. This correlation does not indicate which method is more accurate and representative of the actual bank erosion, which does not advance the stated objectives of this work. The percent difference data (Table 2) shows that the measurement results from the different methodologies are vastly different.

2. There is no effort to consider how channel morphology can influence erosion measurements and catchment sediment calculations (i.e. meander bend locations). Selection of particular river channel morphologies can significantly bias subsequent calculations of catchment loads. It is important to discuss the influence of channel morphology on your site selections and measurement results. Henshaw et al. (2012) and Kessler et al. (2013) may provide a good jumping off point. See also Kimiaghalam et al. (2015)

3. I think the real value of this work is how the comparison and evaluation of these different methodologies helps to resolve some of the complexity and challenges inherent to measuring streambank erosion and deriving catchment sediment loads. The discussion seems to focus on the comparison of methods but not on how well each methodology captures the nature of streambank erosion in the catchment. This is key to justify why the erosion pin data alone was chosen to derive catchment sediment load values.

4. This paper can be written much more clearly. I would suggest that the author(s) comb through each paragraph to identify redundant or wordy phrasing to tighten up and clarify the message.

References:

Henshaw, A.J., Thorne, C.R., Clifford, N.J., 2012. Identifying causes and controls of river bank erosion in a British upland catchment. CATENA, 100(0), 107-119.

Kessler, A.C., Gupta, S.C., Brown, M.K., 2013. Assessment of river bank erosion in Southern Minnesota rivers post European settlement. Geomorphology, 201(0), 312-322.

Kimiaghalam, N., Goharrokhi, M., Clark, S.P., Ahmari, H., 2015. A comprehensive fluvial geomorphology study of riverbank erosion on the Red River in Winnipeg, Manitoba, Canada. Journal of Hydrology, 529, 1488-1498.

Author Response

Reviewer 2 Response (water-571000):

This paper compares different streambank erosion measurement techniques in an effort to highlight the utility of different measurement methodologies (erosion pins, total station, and TLS) across a range of bank conditions. The authors seek to not only compare the data gathered from each method but to also evaluate each method in terms of how well it represents the streambank changes over time. The authors then use their erosion measurements to derive a catchment sediment load attributable to streambank erosion.

The fundamental field surveys and laboratory analyses are sound; however, there are several issues with how the results are presented and discussed. Additionally, the writing of this manuscript, while mostly grammatically correct, contains numerous redundancies, overly verbose, and unclear phrasing that detracts from the message that the authors seek to convey. A few general issues are listed below, and I refer you to the manuscript file for in-text comments. Response: We thank the reviewer for taking the time to make a thorough review of our manuscript. The revisions and resources provided have been very beneficial. They helped us to increase the clarity of the writing and more effectively explain and discuss our results. The reviewer’s suggestions have certainly led to a more readable and impactful paper. We placed most of our responses in the in-text manuscript PDF file (attached) as replies to the reviewer’s numerous individual comments. We also noted changes in red in the revised manuscript file showing edits.

The language describing the ANOVA and individual correlation tests is unclear. The authors use the word ‘differences’ in place of ‘correlation’, and they use ‘correlation’ to indicate similarity in the data. The axes of the graphs in Figure 2 demonstrate this point. A statistically significant correlation was found between the total station and TLS data (Figure 2 c). However, the data from these two measurement methods are an order of magnitude off, as seen in the range of the axes. This correlation is only valuable if you are trying to predict the results of one method by employing the other. This correlation does not indicate which method is more accurate and representative of the actual bank erosion, which does not advance the stated objectives of this work. The percent difference data (Table 2) shows that the measurement results from the different methodologies are vastly different. Response: We appreciate the reviewer for pointing out where our writing is unclear and suggesting ways to clarify it. Following the suggestions, we have made numerous changes to the results and discussion to clarify the meanings of our findings. The changes are in the revised manuscript showing edits and the response to the reviewer’s in-text comments. Here we will briefly explain our clarifications. First, we had used the word “differences” to refer to statistical differences for the ANOVA test, but also for absolute or relative differences. We understand how this could be confusing, so we clarified which difference is being referred to for each, using the terms “statistically significant difference,” “absolute error.” and “relative error.” Second, our use of “correlation” was to indicate relationships in the data. We decided to make this clearer by reporting the correlation coefficient R instead of R2 so that the sign of the correlation is reported. We also made clearer that the correlation is valuable when comparing techniques because it aids in finding relationships between the measurements.

There is no effort to consider how channel morphology can influence erosion measurements and catchment sediment calculations (i.e. meander bend locations). Selection of particular river channel morphologies can significantly bias subsequent calculations of catchment loads. It is important to discuss the influence of channel morphology on your site selections and measurement results. Henshaw et al. (2012) and Kessler et al. (2013) may provide a good jumping off point. See also Kimiaghalam et al. (2015) Response: Thank you for this very helpful feedback. We rewrote Section 4.2 to communicate how morphology, fluvial processes, and other factors could affect our findings of the spatial distribution of bank erosion. We also discussed some of the challenges of extrapolating discrete streambank erosion measurements for a basinwide estimation. We included the Henshaw and Kessler references, plus a few others.

I think the real value of this work is how the comparison and evaluation of these different methodologies helps to resolve some of the complexity and challenges inherent to measuring streambank erosion and deriving catchment sediment loads. The discussion seems to focus on the comparison of methods but not on how well each methodology captures the nature of streambank erosion in the catchment. This is key to justify why the erosion pin data alone was chosen to derive catchment sediment load values. Response: Thank you for these good suggestions. We rewrote much of the discussion to incorporate these improvements. In Section 4.1, we added a final paragraph which discusses a potential use of different techniques together (albeit for different purposes within a study) to reduce the complexities and challenges of streambank erosion studies. We also rewrote the first paragraph of Section 4.3 to discuss these challenges, incorporating the limitations and errors of the techniques. Here, we also added a reference to Section 2.7, which explains our choice of the erosion pins for basinwide estimates because of more sites and no limitations in coverage due to vegetation or other obstructions. This explanation previously was buried and unclear in the discussion. We also rewrote Section 4.2 to discuss challenges of deriving catchment sediment loads from the discrete measurements of our techniques.

This paper can be written much more clearly. I would suggest that the author(s) comb through each paragraph to identify redundant or wordy phrasing to tighten up and clarify the message. Response: We would once again like to thank the reviewer for being immensely helpful for improving the clarity and utility of our findings. We carefully combed through our manuscript and clarified our writing. We have made numerous improvements to the clarity of our manuscript that occur now in nearly every paragraph. Our revised manuscript shows edits for the specific changes we have made.
